# Maturity-Associated Polygenic Profiles of under 12–16-Compared to under 17–23-Year-Old Male English Academy Football Players

**DOI:** 10.3390/genes14071431

**Published:** 2023-07-12

**Authors:** Alexander B. T. McAuley, Ian Varley, Adam J. Herbert, Bruce Suraci, Joseph Baker, Kathryn Johnston, Adam L. Kelly

**Affiliations:** 1Faculty of Health, Education and Life Sciences, Birmingham City University, Birmingham B15 3TN, UK; adam.herbert@bcu.ac.uk (A.J.H.); adam.kelly@bcu.ac.uk (A.L.K.); 2Department of Sport Science, Nottingham Trent University, Nottingham NG11 8NS, UK; ian.varley@ntu.ac.uk; 3Academy Coaching Department, AFC Bournemouth, Bournemouth BH7 7AF, UK; bruce.r.suraci@gmail.com; 4School of Kinesiology and Health Science, York University, Toronto, ON M3J 1P3, Canada; bakerj@yorku.ca (J.B.); krobinso@yorku.ca (K.J.)

**Keywords:** athlete development, genomics, maturation, puberty, soccer

## Abstract

The purpose of this study was to examine polygenic profiles previously associated with maturity timing in male academy football players across different age phases. Thus, 159 male football players from four English academies (U12–16, *n* = 86, aged 13.58 ± 1.58 years; U17–23, *n* = 73, aged 18.07 ± 1.69 years) and 240 male European controls were examined. Polygenic profiles comprised 39 single nucleotide polymorphisms and were analysed using unweighted and weighted total genotype scores (TGSs; TWGSs). There were significant differences in polygenic profiles between groups, whereby U17–23 players had more genetic variants associated with later maturity compared to U12–16 players (TGS, *p* = 0.010; TWGS, *p* = 0.024) and controls (TGS, *p* = 0.038; TWGS, *p* = 0.020). More specifically, U17–23 players had over two-times the odds of possessing >36 later-maturing alleles than <30 compared to U12–16 players (odds ratio (OR) = 2.84) and controls (OR = 2.08). These results suggest there was a greater proportion of relatively later-maturing players as maturation plateaus towards adulthood, which may be explained by the ‘underdog hypothesis’. This study provides the first known molecular evidence that supports the notion that a maturity selection bias exists within male academy football.

## 1. Introduction

Biological maturation is an important variable to consider during youth athletic development in male football (i.e., soccer), as it has been shown to influence player performance capacities and selection/deselection decisions within academy systems [1,2]. Maturation can be defined in an academy football context as a player’s process of progression toward achieving a biologically mature adult state and is often described in terms of status (i.e., stage of maturation), timing (i.e., onset of specific events), and tempo (i.e., rate of changes) [3]. Indeed, there is considerable inter-individual variability between players of the same chronological age, where there can be as much as five–six years variation in the skeletal age (an established index of maturation status in youth) of children with the same chronological age [4,5]. Moreover, it has been reported that between 10.7 and 15.2 years of age, male academy soccer players undergo an accelerated phase of stature growth (i.e., peak height velocity (PHV)) of approximately 7.5 to 9.7 cm a year [6,7].

The variation in maturation is problematic for practitioners given the changes in anthropometric (e.g., stature and lean mass) and physiological (e.g., speed and power) characteristics that accompany earlier maturation are considered important factors during identification and development decision-making processes [8,9,10]. As a consequence, a systematic selection bias towards earlier-maturing players appears to be prevalent across academy football, generally within the under (U)12 to U16 age phase, which coincides with the typical onset of puberty and spans PHV [11,12,13]. Later-maturing players are, therefore, more likely to be excluded from academies, with complete deselection in some instances by U15 [12] and, thus, less represented in senior football.

When considering who achieves expertise at adulthood, however, later-maturing players retained in academy systems may be proportionally more likely to succeed in adulthood [14], possibly due to a phenomenon known as the ‘underdog hypothesis’ [15]. This hypothesis is thought to be explained by later-maturing players experiencing a comparatively greater challenge during early to middle adolescence, developing superior psychological, technical, and tactical skills, which become salient in late adolescence to early adulthood as the advantages associated with advanced maturation become attenuated [11]. Indeed, Cumming et al. [11] found that later-maturing players reported more adaptive engagement in self-regulated learning, in particular self-evaluation and reflection, which may help mitigate some of the physical and functional disadvantages associated with later maturation and provide an athletic advantage in adulthood.

The variation in maturation between individuals is due to a combination of genetic and environmental factors [16,17]. For instance, twin studies have produced heritability estimates ranging from 50% to more than 90% for some maturation phenotypes in both sexes (e.g., menarche age in females and PHV in males) [18,19]. More recent studies have also explored what specific genetic variants may account for a proportion of the estimated heritability in maturation variability [20,21,22]. Most of these studies have used female participants and their menarche age as the marker of maturation due to it being widely recalled and measured, allowing for larger study designs [23]. As an example, a recent large genome-wide association study (GWAS) with 329,345 females identified 389 independent genetic variants associated with age at menarche [21].

In males, voice-breaking age is commonly used as a similar maturation marker, with a GWAS in 55,871 males identifying 14 independent genetic variants associated with age at voice breaking and a high (rg = 0.74) genetic correlation between sexes in menarche and voice-breaking age [20]. Whilst males and females share similar genetic architecture, which displays comparable effect sizes with maturation timing, there are still many genetic variants that differ between sexes and even have directionally discordant allelic effects (i.e., the earlier-maturity allele in one sex is the later-maturity allele in the other) [20]. As such, large sex-specific genetic association studies on maturation are important in order to adequately untangle the complexity of the molecular mechanisms underpinning maturity variation. To address the relative lack and small scale of male-specific studies, Hollis et al. [24] conducted a GWAS on 205,354 males and identified 76 independent single nucleotide polymorphisms (SNPs) associated with maturity timing.

The influence of genetics is a largely under-researched component of athletic development and performance in football but has been growing rapidly [25,26]. Genetic association studies in football have recently investigated what genetic variants may explain some of the inter-individual variability in injury, physiological, psychological, and technical phenotypes, as well as differences in competitive playing levels [27,28,29,30]. A few studies have also explored how differences in chronological and biological age can affect genetic associations with some of these phenotypes [31,32,33]. However, to the authors’ knowledge, no study has yet explored the influence of genetic variants associated with maturation in a football-specific context. Such information may have important implications for future research examining genetic associations within youth and senior football.

The purpose of this study is to examine polygenic profiles (i.e., the combination of several genetic variants) associated with maturity timing in male academy football across different age phases. In light of the current genetic and football research on maturity, we hypothesized that younger age groups would have more genetic variants associated with earlier maturity due to the typical onset of puberty and PHV, alongside the apparent selection bias towards players who have undergone earlier maturity within academy football. In contrast, we predicted that older age groups would have more genetic variants associated with later maturity due to the typical cessation of puberty and PHV, as well as the ‘underdog hypothesis’, whereby later-maturing players are suggested to be proportionally more likely to succeed towards the adult level.

## 2. Materials and Methods

### 2.1. Participants

We examined 159 male football players from four English academies (U12–16, *n* = 86, aged 13.58 ± 1.58 years; U17–23, *n* = 73, aged 18.07 ± 1.69 years), and data from 240 male European controls from the 1000 Genomes database were examined. Informed assent and consent from all players, parents/guardians, and each academy were obtained prior to the commencement of the study. All experimental procedures were conducted in accordance with the guidelines in the Declaration of Helsinki, and ethical approval was granted by the corresponding author’s institutional Ethics Committee.

### 2.2. Genetic Procedures

#### 2.2.1. Genotyping

Procedures were in accordance with previous research and are detailed in McAuley et al. [32]. In brief, following at least 30 min without the ingestion of food and drink, sterile, self-administered buccal swabs were used to collect saliva from players. Saliva samples were sent to AKESOgen, Inc. (Peachtree Corners, GA, USA) within 36 h to extract DNA on an automated Kingfisher FLEX instrument using Qiagen chemistry (Thermo Fisher Scientific, Waltham, MA, USA). The Affymetrix Axiom high-throughput 2.0 protocol was followed throughout, and the Affymetrix Axiom Analysis Suite (Affymetrix, Santa Clara, CA, USA) was used to perform data analysis.

#### 2.2.2. Polymorphism Selection

Of the 76 SNPs associated with male maturity timing in Hollis et al. [24], 39 SNPs (see Table 1) were included in this study due to the coverage of the microarray and following quality control procedures (i.e., minor allele frequency > 0.05 and SNP/sample call rate > 95%).

#### 2.2.3. Polygenic Profiles

To assess differences in polygenic profiles between players and controls, unweighted and weighted total genotype scores (TGSs; TWGSs) were calculated. Each genotype was assigned a score between 0 and 2 based on its association with maturity timing according to Hollis et al. [24] using a co-dominant model (AA vs. Aa vs. aa) (i.e., homozygous genotypes associated with earlier maturity = 2; heterozygotes = 1; alternate homozygous genotypes = 0).

The TGS was calculated following the original Williams and Folland [34] procedure.
TGS = (combined genotype scores/maximum genotype scores) × 100

The TWGS was calculated by multiplying each genotype score by the corresponding SNP effect size (i.e., β) estimates derived from the summary statistics reported by Hollis et al. [24] to create weighted genotype scores.
TWGS = (combined weighted genotype scores/maximum weighted genotype scores) × 100

### 2.3. Data Analysis

Hardy–Weinberg equilibrium (HWE) was assessed using Fisher’s exact tests. Differences between players and controls in polygenic profiles were assessed using linear regression. To estimate effect size, polygenic profiles were split into equal thirds using tertiles, and comparisons between groups were made using odds ratios (ORs) and 95% confidence intervals (CIs). Statistical significance was set at *p* < 0.05. Jamovi version 2.8.23 was used to analyse data.

## 3. Results

All SNPs were in HWE, except for *MIR193B* (rs1659127; *p* = 0.022) and *MKLN1* (rs11763660; *p* = 0.014) in controls. The TGS of groups ranged from 35 to 58 in U12–16 players, 33 to 54 in U17–23 players, and 32 to 58 in controls (see Figure 1). There were significant TGS differences between groups (*F* (2, 396) = 3.48, *p* = 0.032). The mean TGS of U17–23 players (42.5 ± 4.82) was significantly lower than U12–16 players (44.5 ± 5.00; *t* (157) = 2.58, *p* = 0.010) and controls (43.9 ± 5.15; *t* (311) = 2.08, *p* = 0.038). Compared to U17–23 players, U12–16 players and controls had 2.84- and 2.08-times the odds of having a TGS in the higher third (47–58) than in the lower third (32–40), respectively (OR = 2.84, CI: 1.19–6.78; OR = 2.08, CI: 1.00–4.32).

The TWGS of players ranged from 34 to 55 in U12–16 players, 32 to 54 in U17–23 players, and 31 to 57 in controls (see Figure 2). There were significant TWGS differences between groups (*F* (2, 396) = 3.22, *p* = 0.041). The mean TWGS of U17–23 players (41.9 ± 5.09) was significantly lower than U12–16 players (43.7 ± 4.89; *t* (157) = 2.26, *p* = 0.024) and controls (43.5 ± 5.08; *t* (311) = 2.34, *p* = 0.020). Compared to U17–23 players, controls had 2.03-times the odds of having a TWGS in the higher third (46–57) than in the lower third (31–40; OR = 2.03, CI: 1.05–3.91).

## 4. Discussion

This study examined collective differences in the genotype frequency distribution of 39 SNPs associated with maturity timing between U12–16 and U17–23 male English academy football players. To our knowledge, this is the first assessment of a polygenic profile associated with male maturation within a football-specific context. As such, these findings provide initial molecular evidence that supports the suggestion that there may be a maturity selection bias within male academy football, as well as a methodological foundation for future replication studies in this area. Consistent with previous sports genomic research, the findings suggest there is significant genetic variation between male academy football players within the early- to middle-adolescent years and late-adolescent to early-adulthood years. Confirming our hypotheses, the younger age groups had more genetic variants orientated towards earlier maturity, whereas the older age groups had comparatively more genetic variants orientated towards later maturity. Corresponding with non-genetic research in this area, these results suggest that whilst earlier-maturing players are likely more represented during the typical onset of puberty and PHV span, as maturation plateaus towards adulthood, a greater proportion of relatively later-maturing players become better represented.

Genetic variation between male academy footballers of distinct age groups has been noted in earlier studies using SNPs previously associated with physiological, psychological, technical, and injury phenotypes. For instance, Murtagh et al. [33] found that the genotype frequency distribution of *ACTN3* rs1815739, *AGT* rs699, *PPARA* rs4253778, and *NOS3* rs2070744 was significantly different in pre- (aged 10.6 ± 1.4 years) and post- (aged 16.8 ± 2.3 years) peak-height velocity players. Similarly, Hall et al. [31] reported that the genotype frequency distribution of *EMILIN1* (rs2289360) was significantly different between pre- (aged 11.5 ± 1.1 years) and post- (aged 17.5 ± 2.1 years) peak-height velocity players. More recently, McAuley et al. [32] showed that the individual and collective genotype frequency distribution of *IL6* (rs1800795) and 32 other genetic variants was significantly different between U12–16 (aged 13.84 ± 1.63 years) and U17–23 (aged 18.09 ± 1.51 years) players. This genetic variation between players of different chronological and biological ages may be explained by the importance of specific characteristics (i.e., physiological, psychological, technical, and tactical skills) and injury profiles (which have been shown to be influenced by genetic factors [35,36,37]) changing throughout development in youth football [31,32,33,38].

The specific genetic variation observed in this study provides further support for the influence of heritable factors on athlete development and offers molecular evidence that enhances previous maturation research in football. Indeed, this research corresponds with the well-documented disproportionate overrepresentation of relatively earlier-maturing players within male academy football spanning early- to middle-adolescent age groups [11,12,13]. Moreover, these findings are consistent with the ‘underdog hypothesis’, whereby relatively later-maturing players were proportionally more represented towards adulthood, possibly due to the development of superior psychological, technical, and tactical skills [11,15]. Whilst the results of this study suggest the U17–23 group had proportionally more players who were relatively later maturing compared to the U12–16 group, it is important to note that this does not necessarily mean there were more later- than earlier-maturing players in the U17–23 group in absolute terms. In reality, it is likely that due to smaller initial representation within younger age groups because of anthropometric and physiological disadvantages, most later-maturing players may have been excluded or dropped out by the time earlier-maturation advantages become attenuated [12].

The TGS and TWGS were both effective in differentiating U12–16 and U17–23 players, which corresponds with previous research suggesting that each SNP will have a small but additive effect on a given phenotype, as well as each allele having a differing degree of influence [28,29]. Whilst recent genetic research in football using these approaches reported that a TWGS consistently displayed stronger relationships than a TGS [32], this study showed that the TGS was more effective. However, this can be explained by the more literature-driven approach of this study compared to the data-driven approach of previous studies, whereby the allelic directions and sizes of effect in this study were assigned a priori using GWAS summary statistics instead of posteriori based on within-study analyses. An evidence-driven approach was more appropriate for this study due to a large sample of data on maturity timing and the phenotype under investigation being age group. If the allelic directions and sizes of effect were assigned using data based on age-group differences in this study, confounding pleiotropic effects (i.e., influence on other phenotypes) of each SNP may have been introduced instead of solely maturation-based effects.

Importantly, this research expands the knowledge on how genetic variants may affect phenotypes in football-specific contexts. In the future, if genetic information is integrated into athlete development programs, this study will have contributed to a better understanding of the number of phenotypes affected by specific variants. These findings may also have important implications for further studies examining genetic associations in youth and senior football over time due to possible ‘survivor effects’, whereby it is assumed the characteristics of those who prevail in a system reflect the true qualities needed for success [39]. For instance, when examining the genetic profile of athletes at older ages or senior competitive playing levels, consideration should be given to how current selection biases (e.g., advanced maturity) may influence associations and how these may change if improved methodological processes are introduced within development systems. Existing research on athlete development in general is limited with regards to understanding why ‘non-survivors’ leave systems and how this could change if programs were executed differently [40].

At present, the practical implementation of genetic information in football requires the identification of many additional genetic variants and more accurate weightings applied to specific phenotypes. However, recent research indicates that once samples become large enough, polygenic profiles may assist with more accurate estimations of a number of phenotypes applicable to football. For instance, over the past five years, the average sample size of published GWASs has more than tripled, which has led to a substantial increase in the number of significant associations identified and the accuracy of polygenic scores [41]. Using height as an example, the latest study included 5.4 million individuals and identified 12,111 independent associations that explained up to 45% of the variance in populations of European ancestry [42]. This generated a polygenic score that accounted for 23 cm in mean height differences between individuals at the extreme ends (2.5 SD above versus 2.5 SD below the mean) of the distribution [41]. Moreover, the combination of these genetic variants and average parental height produced a significantly better prediction model of height, which had 55.2% accuracy [42]. Once genetic studies in maturation attain samples of a relatively comparative size to those used in height, perhaps the combination of the identified variants could be used to improve current prediction models of maturity in youth football.

The present study is not without some limitations that should be taken into account when interpreting the results. As with previous football genomic studies, despite recruiting players from four academies, as well as comprising a cohort larger than the average in this area (see [25] for a review), the sample size was still relatively small. However, this is a real-world reflection of talent development environments, as many athletes will not possess higher-performance capacities. This also meant that yearly age group comparisons could not be conducted, which may have added greater context to the findings. Building this research base with studies using transparent and consistent methodologies will be important to facilitate research synthesis approaches in the future [43,44]. Whilst we are confident in the findings due to previous research showing strong relationships between the variants investigated and maturity, the actual maturation status of participants was not known, and future research may be improved by including this measurement. Furthermore, the controls used in this study may have introduced population stratification issues for some analyses as they were of European origin and their age was unknown. Future research should aim to recruit age-matched controls from the sample’s country of origin.

It is also important to acknowledge that these findings are only in relation to male players and most likely would not transfer well to a female context, as there are still many variations in genetic associations with maturation between sexes [20]. Future work would benefit from replicating this study within a female sample to better understand sex-specific developmental programming. Common genetic variants such as SNPs appear to contribute more to earlier maturity in females compared to males [21], which suggests a similar study may have greater power to detect between-group differences in female players. Whilst the number of genetic variants incorporated in this study appears to be the most used within a football-specific context, only SNPs were genotyped, and potential epistasis (i.e., gene–gene interactions) was not considered. Many other types of genetic variations exist (e.g., copy number variants and insertions–deletions), and the interactions between them could alter the association of polygenic profiles. The addition of more SNPs and other genetic variants, as well as considering their interactions, may increase polygenic profile accuracy in the future. Lastly, *MIR193B* (rs1659127) and *MKLN1* (rs11763660) deviated from HWE, which may be due to not correcting for family-wise error rates that were inflated due to the 39 comparisons. However, genotyping error may also have occurred and is an important consideration when interpretating the findings.

## 5. Conclusions

This study presents novel molecular evidence with regards to the collective genotype frequency distribution of 39 SNPs associated with the maturity timing of male academy football players in England. The key findings showed that there was significant genetic variation between U12–16 and U17–23 players, whereby U17–23 players had a polygenic profile comprising more alleles associated with relatively later maturity compared to U12–16 players. These results support previous research that suggested there may be a maturity selection bias within male academy football that favours earlier-maturing players during early to middle adolescence, but as maturation plateaus towards adulthood, a greater proportion of relatively later-maturing players become better represented. It is important that these findings are replicated in larger independent samples of youth football players, as well as the identification of many additional genetic associations with maturity before practical applications are contemplated.

## Figures and Tables

**Figure 1 genes-14-01431-f001:**
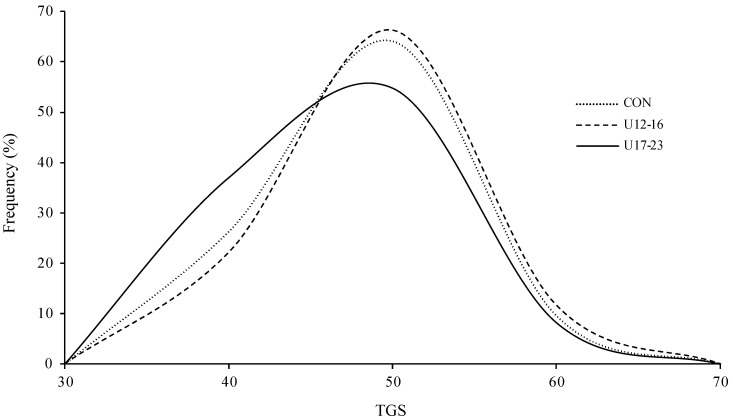
Total genotype score (TGS) frequency distributions in under (U)12 to 16 and U17 to 23 male academy football players, as well as male European controls (CON).

**Figure 2 genes-14-01431-f002:**
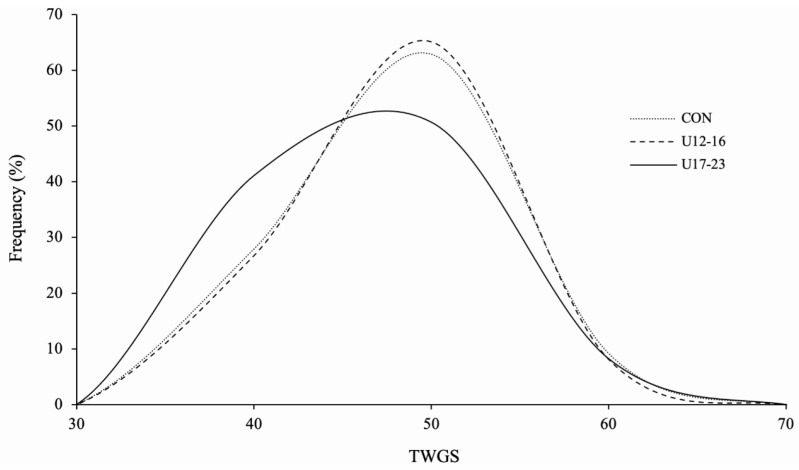
Total weighted genotype score (TWGS) frequency distributions of in under (U)12 to 16 and U17 to 23 male academy football players, as well as male European controls (CON).

**Table 1 genes-14-01431-t001:** Effect size and allele frequency of single nucleotide polymorphisms (SNPs).

Gene	SNP	β	Minor Allele Frequency %		
			Controls	U12–16	U17–23
*ZNF483*	rs10980922	0.055	7	9	7
*HERC2*	rs7402990	0.051	10	8	11
*C16orf55*	rs35063026	0.051	7	9	10
*MIR193B*	rs1659127	0.050	34	30	25
*LIN28B*	rs11156429	0.049	45	26	26
*TMEM38B*	rs9408817	0.041	32	33	33
*SOX2OT*	rs73182377	0.040	28	26	32
*C11orf63*	rs6589961	0.038	43	41	39
*PHF15*	rs62379978	0.036	16	14	17
*C1orf127*	rs77578010	0.036	21	26	23
*MKLN1*	rs71578952	0.035	48	41	48
*IRF4*	rs12203592	0.035	12	19	12
*NCOA6*	rs4911442	0.029	10	17	13
*PRDM2*	rs2473234	0.029	15	11	18
*DET1*	rs17190166	0.028	42	47	43
*AKAP1*	rs17833789	0.028	44	40	46
*MIR548A1*	rs2842385	0.027	16	16	22
*BDNF*	rs2049045	0.027	21	20	16
*ZNF536*	rs11671893	0.026	18	12	14
*TMEM18*	rs10188334	0.025	19	17	22
*SEC16B*	rs6670873	0.025	20	17	16
*PDGFA*	rs9690350	0.025	41	42	49
*CYFIP2*	rs438830	0.025	21	24	14
*RORA*	rs3743266	0.024	36	32	36
*ODZ2*	rs2923177	0.023	45	48	48
*SATB2*	rs1598656	0.022	27	31	23
*PRICKLE4*	rs6925777	0.022	49	49	47
*RMI1*	rs7853970	0.022	47	42	45
*HPGDS*	rs767657	0.022	40	40	35
*FTO*	rs1121980	0.022	47	45	46
*FPGT-TNNI3K*	rs1514177	0.022	44	42	49
*IGSF11*	rs10934420	0.020	49	47	47
*ADARB2*	rs7896371	0.020	43	47	41
*GCKR*	rs780094	0.020	41	32	35
*HLF*	rs12940636	0.020	29	33	35
*ZNF324B*	rs4801593	0.020	27	26	28
*DLGAP1*	rs11873906	0.020	27	31	28
*FAM178A*	rs11190751	0.020	44	42	49
*KDM4C*	rs913588	0.019	44	42	47

## Data Availability

The data presented in this study are available on request from the corresponding author.

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
