# Peer review of "Maturity-Associated Polygenic Profiles of under 12–16-Compared to under 17–23-Year-Old Male English Academy Football Players"

_genes, 2023, doi:10.3390/genes14071431_

Round 1
Reviewer 1 Report
This study is well planned and performed. The authors have produced a lot of interesting data which deserves publication and I have no major criticisms. Nevertheless, the paper could be improved, please see my comments and recommendations below:
Minor Comments:
1. At the end of the introduction section, together with the hypothesis, the purpose of the study must be presented.
2. In the paper (Line 203-204) please indicate correctly the gene names (should be written in Italic).
3. Although the authors indicated that the study has no limitations, the age of the male in the control group is not known. Could this have influenced the results of the study?
Aside from these concerns, the authors should be congratulated on performing a genetic study in football players of sufficient number and combining hypothesis based polygenic profiles - much needed in this field.
Author Response
REVIEWER: This study is well planned and performed. The authors have produced a lot of interesting data which deserves publication and I have no major criticisms. Nevertheless, the paper could be improved, please see my comments and recommendations below:
AUTHORS: Thank you for taking the time to review our paper.
REVIEWER: At the end of the introduction section, together with the hypothesis, the purpose of the study must be presented.
AUTHORS: Thank you for your comment. We do briefly state the purpose of the study at the end of the paragraph before the hypothesis but acknowledge this may be better placed with the hypothesis and extended more. This has now been amended.
REVIEWER: In the paper (Line 203-204) please indicate correctly the gene names (should be written in Italic).
AUTHORS: Thank you for highlighting these errors. These have now been corrected.
REVIEWER: Although the authors indicated that the study has no limitations, the age of the male in the control group is not known. Could this have influenced the results of the study?
AUTHORS: Thanks for your comment. Age-matched controls from England would have been ideal but unfortunately this was not possible and the age of 1000 genomes project samples was not available. It is not clear how age would have influenced results as we were not directly assessing maturation status but we agree that it is an important point to consider and have now added this to the limitations.
REVIEWER: Aside from these concerns, the authors should be congratulated on performing a genetic study in football players of sufficient number and combining hypothesis based polygenic profiles - much needed in this field.
AUTHORS: Thanks again for your constructive and positive feedback.
Reviewer 2 Report
Interesting article, easy to read and understand. It generates new knowledge about something little studied: maturity-associated polygenic profiles in jonger football players. It is a complex and well-argued investigation.
Author Response
REVIEWER: Interesting article, easy to read and understand. It generates new knowledge about something little studied: maturity-associated polygenic profiles in jonger football players. It is a complex and well-argued investigation.
AUTHORS: Thank you for taking the time to review our paper.
Reviewer 3 Report
The purpose of the study was to examine polygenic profiles (i.e., the combination of several genetic variants associated with maturity timing) between fifty-nine male football players from English academies (U12–16, n = 86, aged 13.58 ± 1.58 years; U17–23, n = 73, aged 18.07 ± 1.69 years) and data from 240 male European controls. The main limitation (from my point of view) was the low number of athletes per category (< 90). This low number of samples may interfere with the interpretation of results, as presented at the end of the discussion. An effort to increase this cohort is most welcome in future studies.
I congratulate the authors for the originality of the study. Overall, the manuscript is interesting and original, but I have some methodological and prospective (interpretation of results) issues, as follows:
I understood that the control group consisted of 240 European samples retrieved from the 1000 genomes project. I would like to know if these 240 samples are from Europeans of English origin? Was this control group a good reference for the players?
It was mentioned that: Of the 76 SNPs associated with male maturity timing in Hollis et al. [24], 39 SNPs were included in this study due to the coverage of the microarray and following quality control procedures. Does this mean that the chip used contained only the 39 SNPs included in the study?
It was mentioned that: All SNPs were in HWE except for MIR193B (rs1659127; p = 0.022) and MKLN1 (rs11763660; p = 0.014) in controls. Were these SNPs excluded from the analysis (genotype score)?
It was found that younger age groups had more genetic variants orientated towards earlier maturity, whereas the older age groups had comparatively more genetic variants orientated towards later maturity. I have a personal question about these findings: Were the players of the older group not athletes in the younger category some time ago? In this case, if the analyzes were carried out in the past, the findings could be different. Also, do younger athletes (with more genetic variants orientated towards earlier maturity) interrupt their careers earlier? Is there any information about this?
I understand the difficulty, but having a marker of players' maturation status would help strengthen the association. Also, is there any data on whether these players (with different genetic predispositions to biological maturation) perform better within their age categories?
Author Response
REVIEWER: The purpose of the study was to examine polygenic profiles (i.e., the combination of several genetic variants associated with maturity timing) between fifty-nine male football players from English academies (U12–16, n = 86, aged 13.58 ± 1.58 years; U17–23, n = 73, aged 18.07 ± 1.69 years) and data from 240 male European controls. The main limitation (from my point of view) was the low number of athletes per category (< 90). This low number of samples may interfere with the interpretation of results, as presented at the end of the discussion. An effort to increase this cohort is most welcome in future studies.
I congratulate the authors for the originality of the study. Overall, the manuscript is interesting and original, but I have some methodological and prospective (interpretation of results) issues, as follows:
AUTHORS: Thank you for taking the time to review our paper.
REVIEWER: I understood that the control group consisted of 240 European samples retrieved from the 1000 genomes project. I would like to know if these 240 samples are from Europeans of English origin? Was this control group a good reference for the players?
AUTHORS: Thanks for your comment. The 240 samples are from European origin. Age-matched controls from England would have been ideal but unfortunately this was not possible and the age of 1000 genomes project samples was not available. We agree that it is an important point to consider and have now added this to the limitations.
REVIEWER: It was mentioned that: Of the 76 SNPs associated with male maturity timing in Hollis et al. [24], 39 SNPs were included in this study due to the coverage of the microarray and following quality control procedures. Does this mean that the chip used contained only the 39 SNPs included in the study?
AUTHORS: The chip used contained more than the 39 SNPs but they did not pass the quality control procedures (i.e., minor allele frequency >0.05 and SNP/sample call rate >95%).
REVIEWER: It was mentioned that: All SNPs were in HWE except for MIR193B (rs1659127; p = 0.022) and MKLN1 (rs11763660; p = 0.014) in controls. Were these SNPs excluded from the analysis (genotype score)?
AUTHORS: These SNPs were retained as this may have been due to a wide range of reasons such as not correcting for Family-Wise Error Rates that were inflated due to the 39 comparisons. However, as deviation from HWE can be due to genotyping error, we have now added this as a limitation towards the end of the discussion.
REVIEWER: It was found that younger age groups had more genetic variants orientated towards earlier maturity, whereas the older age groups had comparatively more genetic variants orientated towards later maturity. I have a personal question about these findings: Were the players of the older group not athletes in the younger category some time ago? In this case, if the analyzes were carried out in the past, the findings could be different. Also, do younger athletes (with more genetic variants orientated towards earlier maturity) interrupt their careers earlier? Is there any information about this?
AUTHORS: Yes, the older players in this sample would have been in the younger category a few years ago but they would have been accompanied by additional players who may have been earlier maturers. To your second point, non-genetic data indicates athletes who mature earlier have an advantage in younger age groups but lose this advantage in older age groups as maturation plateaus and are more likely to be deselected. As for genetic data, to our knowledge this is the first study to investigate this and supports non-genetic findings.
REVIEWER: I understand the difficulty, but having a marker of players' maturation status would help strengthen the association. Also, is there any data on whether these players (with different genetic predispositions to biological maturation) perform better within their age categories?
AUTHORS: We completely agree and have mentioned in the limitations that the actual maturation status of participants were not known and future research may be improved by including these measurements. To your second point, we are not aware of any studies that have investigated the association between performance and maturity-associated genetic variants. Non-genetic data has shown consistent associations between maturity and physiological capacities though.
Round 2
Reviewer 3 Report
The authors answered all questions. I have no additional comments.